# P-Loop Channels: Experimental Structures, and Physics-Based and Neural Networks-Based Models

**DOI:** 10.3390/membranes12020229

**Published:** 2022-02-16

**Authors:** Denis B. Tikhonov, Boris S. Zhorov

**Affiliations:** I.M. Sechenov Institute of Evolutionary Physiology and Biochemistry, Russian Academy of Sciences, 194223 St. Petersburg, Russia; boriszhorov@yahoo.com

**Keywords:** ligand–channel interactions, sequence alignment, π-bulges, crystal structures, cryo-EM structures, potassium channels, sodium channels, calcium channels, TRP channels, ionotropic glutamate receptors

## Abstract

The superfamily of P-loop channels includes potassium, sodium, and calcium channels, as well as TRP channels and ionotropic glutamate receptors. A rapidly increasing number of crystal and cryo-EM structures have revealed conserved and variable elements of the channel structures. Intriguing differences are seen in transmembrane helices of channels, which may include π-helical bulges. The bulges reorient residues in the helices and thus strongly affect their intersegment contacts and patterns of ligand-sensing residues. Comparison of the experimental structures suggests that some π-bulges are dynamic: they may appear and disappear upon channel gating and ligand binding. The AlphaFold2 models represent a recent breakthrough in the computational prediction of protein structures. We compared some crystal and cryo-EM structures of P-loop channels with respective AlphaFold2 models. Folding of the regions, which are resolved experimentally, is generally similar to that predicted in the AlphaFold2 models. The models also reproduce some subtle but significant differences between various P-loop channels. However, patterns of π-bulges do not necessarily coincide in the experimental and AlphaFold2 structures. Given the importance of dynamic π-bulges, further studies involving experimental and theoretical approaches are necessary to understand the cause of the discrepancy.

## 1. Introduction

Among the various families of ion channels, tetrameric P-loop channels stand alone due to their large functional diversity and importance in physiology, pathophysiology, pharmacology, and toxicology [1,2,3,4,5,6]. This superfamily includes various voltage- and ligand-gated potassium channels, voltage-gated sodium and calcium channels, TRP (transient receptor potential) channels, and ionotropic glutamate receptors [7,8,9,10]. The common structural motif of all the family members is the pore-forming domain (PD). The latter contains the outer and inner transmembrane helices linked by a membrane re-entrant P-loop with a membrane-descending pore helix (P-helix). The C-ends of the four P-helices with few residues at the C-end converge to the pore axis to form a pore that narrows with the selectivity filter (SF) (Figure 1). The activation mechanisms of P-loop channels are very different due to specific structural and functional properties of the gating–control domains, which are attached to the pore domain [11,12,13,14,15,16]. 

The P-loop channels include tetramers (glutamate receptors, TRP, and potassium channels), dimers of dimers (two-pore channels) [17], and pseudo-tetramers. The latter are eukaryotic sodium and calcium channels in which large pore-forming subunit folds from a single polypeptide chain of four homologous repeats [1,2]. Interestingly, even the orientation of P-loops in the membrane is not conserved in P-loop channels—in ionotropic glutamate receptors, the P-loops are partially exposed to cytoplasm, whereas in other members of the superfamily, the P-loops are partially exposed to the extracellular space [18,19].

Figure 1A–C shows the organization of eukaryotic voltage-gated sodium and calcium channels according to the structural data. The alpha subunit is assembled from homologous repeats I–IV. Each repeat has the VSD, which includes transmembrane segments S1–S4, and contributes a quarter to the PD (segments S5, P, and S6). The PD surrounds the central pore, whereas VSDs are localized peripherally (Figure 1B,C). 

The similar architecture of the PD in P-loop channels was recognized before the crystal structures had become available. Location of the activation gate at the cytoplasmic part of the pore [20], location of the selectivity filter at the P-loop turn [21], and the clockwise arrangement of repeats I, II, III, and IV in the sodium channels [22] were determined by experimental studies, which included mutagenesis, electrophysiology, and analysis of drug and toxin action. 

P-loop channels are intensively studied due to their key roles in multiple physiological and pathophysiological processes and their importance in pharmacology and toxicology. Various characteristics of P-loop channels, including 3D structures, ion permeation and gating, impact of disease mutations, and modulation by drugs and toxins are described in many recent reviews, e.g., [2,3,7,8,9,10]. The vast majority of the reviews are focused on specific channel subfamilies, and they only rarely provide comparison with channels from other subfamilies. This is not surprising given the huge functional diversity of the P-loop channels and the distinct roles of individual subfamilies in physiology, pathophysiology, pharmacology, and toxicology. The similar architecture of the pore-forming domain of different P-loop channels calls for a comparative analysis of typical representatives from different channel subfamilies. Such comparison, which may provide novel structural and functional insights, is a major goal of the current review.

We describe crystal and cryo-EM (cryo-electron microscopy) structures and physics-based models of various P-loop channels. We further compare the experimental structures of several channels and their recent AlphaFold2 models. The comparison reveals generally similar folding, but also differences in some important details. In particular, the inner helices of many P-loop channels have π-bulges, which appear and disappear depending on the channel’s functional state and even on the presence of ligands. The patterns of π-bulges do not always coincide in experimental structures and AlphaFold2 models.

## 2. Comparative Structural Analysis of P-Loop Channels

To facilitate comparison of various P-loop channels, we use here a modified labeling scheme, which is universal for P-loop channels [18]. The most sequentially conserved and functionally important residues in transmembrane helices S1–S6 and in P-loops are assigned the number 50 (Figure 1D). A residue label includes the segment designation and a relative residue number in the segment. The C-terminal part of the outer helices (S5) has a small residue (G, A, or S), which is involved in a knob-into-hole contact with a bulky residue at the N-end of P-helix of the same subunit/repeat [23]. The small residue is labeled S5/50. For example, in the Cav1.2 channel, alanine in the first repeat of the outer helix (IS5) is labeled A^IS5/50^ (Figure 1D). Label P/50 is assigned to the selectivity-filter residues in P-loops, which include valine in the TVGYG fingerprint sequence of potassium channels [24], DEKA ring in eukaryotic sodium channels [21], and EEEE ring in eukaryotic calcium channels [25]. The gating hinge glycine in the inner helix of potassium channels [26] and sequentially matching residues in other channels are labeled S6/50. It should be noted that the sequence alignment shown in Figure 2D does not necessarily match alignments proposed by web servers, e.g., Clustal Omega [27].

In order to obtain three-dimensional (3D) alignment P-loop channels, we used as a template the chimeric potassium channel Kv1.2/Kv2.1 (2r9r), the first eukaryotic P-loop channel whose crystal structure was obtained with the resolution below 2.5 Å [28]. The channel was oriented so that the pore axis coincided with the *z*-axis, the tyrosine backbone CA atoms in selectivity filter GYG motif laid in plane *xOz*, and axis *Ox* was directed towards the CA atom of the GYG tyrosine in subunit III. Other P-loop channels were 3D-aligned with the Kv1.2/Kv2.1 channel by minimizing RMS deviations of the CA atoms in positions p38–p47 of four P helices, which are the most 3D-conserved segments of P-loop channels [29]. 

The list of structures used in the study is given in Table 1. The original structures are downloaded from the protein data bank (PDB) (www.rcsb.org, accessed on 25 December 2021) and the AlphaFold server (www.alphafold.ebi.ac.uk, accessed on 25 December 2021).

## 3. Crystal and Cryo-EM Structures

The KcsA potassium channel from the soil bacterium *Streptomyces lividans* was the first P-loop channel whose 3D structure was obtained with X-ray crystallography [26]. The crystal structures of homotetrameric potassium channels served as templates for homology modeling of eukaryotic sodium, calcium, and glutamate receptor channels [73,74,75,76,77,78,79,80]. The similarity between potassium channels and other P-loop channels was often implied to rationalize experimental data (e.g., mutational analysis and drug action), even without building homology models [81,82,83]. 

In agreement with an earlier concept of the open-gate architecture, which was based on indirect experimental data, the crystal structure of the calcium-gated potassium channel MthK revealed diverging C-parts of the inner helices exposed to the intracellular vestibule [84]. Although the open Kv1.2 channel has different conformation of the S6 helices [85], the principal gating mechanism is the same. In both channels, the opening involves shifting and twisting C-terminal parts of the S6 helices due to conformational changes at the conserved gating-hinge glycine residues. Comparison of the open-gate structure of the mammalian Kv1.2 channel and closed-gate structure of a bacterial cyclic nucleotide-regulated channel MlotiK1 [33] revealed the critical role of the S4–S5 linker helices that form a cuff around the bundle of S6 helices and limit their movement during gating. 

The activation gating mechanism involving diverging inner helices is common for P-loop channels. Specific examples are available for almost every member of the superfamily. It was further demonstrated that the selectivity-filter region of the KcsA channel operates as the C-type slow-inactivation gate [86,87]. Rearrangements at the selectivity-filter region that can prevent the ion flow were also reported for other P-loop channels, including Kv7.1 [88], TRPV1 [48], Nav1.4 [89], Cav1.2 [90], and glutamate receptor [91] channels.

Over the past two decades, numerous experimental structures of various P-loop channels have been published (Table 1 and references therein). The structures have confirmed the conserved general architecture of various channels. Figure 2 shows several segments in the 3D-aligned experimental structures of nineteen P-loop channels, which represent potassium, sodium, and calcium channels, as well as TRP channels, two-pore channels, and ionotropic glutamate receptors. The channels have different functional properties, ion selectivity, gating mechanisms, and length and folding of intra- and extracellular loops. Despite these differences, the positions and orientation of the CA–CB bonds in sequentially matching positions of P-loops and S6 helices are very similar (Figure 2B). While the structural similarity of S1 helices in the voltage-sensing domain is less obvious, the positions and orientations of the CA–CB bonds in residues S1/50 are rather similar (Figure 2C). This overall similarity validates the comparative structural analysis of different P-loop channels in different functional states, despite the sequence alignment not always being obvious (Figure 1D).

The experimental 3D structures have shed light on the common and specific features of different P-loop channels and greatly advanced our understanding of the mechanisms of ion permeation, selectivity, gating, and sensitivity to drugs and toxins [3,9,92,93]. The vast majority of experimental structures show P-loop channels in the energetically most-preferred states, with the pore domain in a presumably inactivated state and VSDs in the activated states, e.g., [58]. Notable exceptions include the cryo-EM structures of eukaryotic sodium channels with the open pore domain [59] or with VSDs deactivated by proteinaceous toxins [61,65]. The crystal structures of the NavAb channel, in which engineered disulfide bridges stabilize the pore domain in the closed or open states [43] or VSDs in the deactivated state [45], provide important templates for modeling eukaryotic channels in sparsely populated, but functionally important conformations. These structures were used, in particular, to build models of the medically important channels hCav1.2 and hNav1.5 in different states, to map state-dependent contacts, and to suggest the structural mechanisms of dysfunction of the channels’ disease variants [94,95,96]. 

The available 3D structures have opened an avenue for computational studies that have addressed important problems, which currently cannot be resolved experimentally. Examples include MD simulations of ion permeation through prokaryotic potassium channels [97,98,99], MD simulation of voltage-sensing gating with the supercomputer Anton [100], and computational studies of ion selectivity in potassium [101,102,103] and sodium [104,105,106,107] channels.

Besides the conserved structural and functional features of P-loop channels, experimental structures have revealed important differences between different channel types. The crystal structures of the prokaryotic sodium channels NavAb with the closed pore [41], NavMs with the open pore [47], and NavRh in a presumably inactivated state [49] revealed their differences with potassium channels not only in the selectivity-filter region, but also in the pore domain architecture. Particularly, while the P-loop of potassium channels has only one helix (a membrane-descending P-helix), P-loops on sodium and calcium channels have two helices separated by the selectivity-filter region: a membrane-descending P1-helix at the N-terminal part of the P-loop and a membrane-ascending helix P2. Furthermore, subunit interfaces in sodium channels are significantly wider than those in potassium channels. 

Experimental 3D structures have led to the significant modification of some structural concepts on P-loop channels. Thus, the ball-and-chain mechanism of the fast (N-type) inactivation of voltage-gated potassium and sodium channels was proposed to involve a short fast-inactivated particle at the channel cytoplasmic part to enter the open pore and physically occlude the ion permeation pathway [108,109,110]. However, the cryo-EM structures of the electric eel channel Nav1.4 [56] and other eukaryotic sodium channels show that the fast-inactivation LFM (IFM) motif in the intracellular linker between repeats III and IV does not block the pore, but binds between helices IIIS4-S5 and IIIS5 and the membrane-oriented faces of helices IIIS6 and IVS6. Another example is the 3D structures of glutamate receptor channels, wherein early X-ray structures suggested that despite the general overall similarity, glutamate-gated channels differ significantly from potassium channels. However, recent studies, which employed improved experimental methods, demonstrated a high similarity between these subfamilies of P-loop channels [111]. 

## 4. Structures of P-Loop Channels with Drugs and Toxins

P-loop channels are targets for numerous naturally occurring toxins [112], medically important and illicit drugs [113,114], and insecticides [115]. Theoretical and experimental studies greatly advanced our understanding of the ion channels’ sensitivity to various ligands. Before the experimental structures of ligand–channel complexes became available, homology models based on the crystal structures of potassium channels and ligand docking methods were used to rationalize, in structural terms, large experimental data accumulated over decades of intensive research. These computational studies employed data from mutational, electrophysiological, and ligand-binding experiments, which determined ligand-binding binding regions and key ligand–channel contacts, e.g., [75,116]. The next generation of more precise ligand–channel models is based on crystal and cryo-EM structures of corresponding channels or their close homologs. Such models have been used not only to rationalize the available experimental data, but also to predict the structures of new ligands, e.g., [117,118]. 

The experimental structures of ligand–channel complexes have allowed for estimating the predictive power of early models. For example, homology models of the NMDA (N-methyl-d-aspartate) and AMPA (α-amino-3-hydroxy-5-methyl-4-isoxazolepropionic acid) glutamate receptor channels with ligands were published long before the first experimental structures of the respective complexes had become available. The experimental structures confirmed major aspects of the model-based predictions, as seen in [111] and the references therein.

Drug binding to different regions of sodium channels was addressed in many studies. Initially, homology models based on the crystal structures of potassium channels were used to dock important drugs, including local anesthetics and other inner pore blockers [75,76]. Small molecular ligands were proposed to reach the binding site in the inner pore through the interface between neighboring S6 helices [119,120]. These models provided the first structural visualization for the hypothesis that local anesthetics and other small molecule drugs can reach the inner pore of the closed sodium channel through a hydrophobic access pathway [121]. Later, the crystal structure of the NavAb channel demonstrated that fenestrations between the S6 helices in the sodium channel are much wider than those in potassium channels [41]. Experimental structures of sodium channels were used to elaborate a new generation of models to explain the action of local anesthetics and related drugs [122,123,124,125]. Recent crystal and cryo-EM structures of sodium channels with drugs in the inner pore [58,63] are consistent with the earlier proposed models of ligand–channel interactions. 

Naturally occurring toxins, including tetrodotoxin, saxitoxin, and mu-conotoxins, which bind to the outer pores of sodium channels, have been used to map their binding sites and understand the basic features of the toxin–channel interactions. Early models, which employed structure–activity data on the toxins and mutational analysis of the channels, predicted the binding sites of the toxins and their orientation in the Nav1.4 channel [76,126,127]. Following publication of the NavAb channel crystal structure [41], more accurate models of toxin-bound channels have been elaborated [29,116,128,129,130]. Recent cryo-EM structures of toxin-bound eukaryotic sodium channels demonstrated both achievements and limitations of the homology models [51,53,64]. 

Drugs that target L-type calcium channels, including phenylalkylamines, benzothiazepines, and dihydropyridines, are used to treat cardiovascular diseases [131]. Previous studies identified amino acid residues whose mutations affect the action of these drugs [82,132]. These data were used to build drug-bound models of the Cav1.2 channel [74,80,133,134,135,136]. The cryo-EM structure of the calcium channel Cav1.1 highlighted the DHP (dihydropyridines) binding region [34], which was previously revealed in intensive mutational studies and visualized in homology models.

Despite the impressive progress in the structural biology of P-loop channels, some problems remain unresolved. Importantly, the static crystal and cryo-EM structures correspond to the lowest-energy state. Thus, mechanisms of state-dependent drug binding and the impact of ligands on the channel transitions between functional states remain matters of speculation. For example, the cryo-EM structures of the L-type calcium channel with DHP agonists and antagonists are virtually the same and do not explain the principally different action of these ligands [34]. Another limitation of experimental structures is that ions, water molecules, lipids, and detergent molecules, which are not always resolved, may affect ligand binding poses. Therefore, the physiological relevance of the experimental structures is not unquestionable. An example is the cryo-EM structure of the L-type calcium channel with verapamil [32], where lipids and detergent molecules strongly interact with the drug and can affect the drug–channel structure.

## 5. π-Bulges in the Inner Helices

Atomic-scale structures have revealed some interesting features of channels that were not considered before. Thus, the inner helices in some channels are not entirely alpha helical, but they contain π-helical elements. An extra residue per helical turn in a π-bulge causes the reorientation of upstream or downstream residues by about 90 degrees as compared to the classical alpha helix. This, in turn, dramatically changes the pattern of inter-segment contacts involving the reoriented part of the helix and alters the pattern of pore-facing residues and ligand–channel contacts. For example, insertion of exceptionally conserved asparagine residues in the inner helices of sodium and calcium channels, which form state-dependent inter-segment H-bonds and stabilize the open conformation of the pore domain [137,138], may explain the evolutionary appearance of π-bulges in these channels, as well as in TRP channels. Accommodation of the additional residue would reorient the C-terminal parts of the inner helixes, which contain the activation gate residues and other residues involved in important inter-segment contact. The π-bulges preserve the orientation of residues beyond the π-helical segment and their intersegment contacts and thus provide structural tolerance to the insertions [139]. 

Different structures of the same P-loop channel may have different π-bulges, suggesting their dynamic nature. Some structures demonstrate structural rearrangements due to π-bulges, which are apparently induced by ligand binding. Examples include the hCav3.1 channel in the apo-state (6kzo) and in the complex with specific T-type channel blocker Z944 (6kzp), as well as rbCav1.1 channel structures with verapamil (6jpa) or diltiazem (6jpb). Besides affecting ligand–channel interactions, π-bulges may substantially change contacts between the S6, S5, and S4–S5 helices [139], and thus change the structural stability of the pore domain and transitions between the channel functional states.

Inner helices in TRP channels show a large diversity of π-bulges [140,141]. Inner helices in the same-type TRPV3 channels either lack π-bulges (6dvy, 6mhw, 6mho, 6pvl, 6dvw, and 6uw9) or have them (6lgp, 6uw4, 6vpo, and 6mhs). Such variations imply that dynamic π-bulges may govern conformational rearrangements of the inner helices. Indeed, comparison of the TRPM6 channel structures indicates that the S6 segments undergo a conformational transition from the α-helix to π-helix upon the channel opening [70]. In TRPV1 and TRPV2 channels, α-helical S6s and S6 helices with energetically less favorable π-bulges may represent different functional states of the channels [50]. Class II and Class III structures of the channel rbCav1.1 indicate that transition of the IIIS6 helix from conformation with the π-helical elements to the alpha-helical conformation is associated the outward motion and axial rotation of the helix [32]. Vanishing π-bulges are associated with the activation gate widening in Cav1.1, but the gate narrowing in TRPV6. Thus, clear relations between the activated gate dimensions and presence of π-bulges are lacking. The π-bulges may govern the bending of S6 helices in the channels without the glycine gating hinges, which are present in potassium channels. 

State-dependent and drug-induced π-bulges, which are seen in some cryo-EM structures, suggest an unusual mechanism by which ligands may affect the channel gating. For example, the sodium channel activators batrachotoxin and veratridine and the DHP agonists and antagonists of the L-type calcium channels change probabilities of the open and closed channel states [82,142,143,144,145]. Cryo-EM structures of the Cav1.1 channel show DHP agonists (7jpk) and antagonists (7jpw) bound in the III/IV fenestration [34], but changes in the activation gate region are very small and do not explain the principle mechanism of action of these important ligands. Drug-induced π-bulges could reorient the S6 residues and thus affect stabilities of the open- and closed-gate conformations. 

## 6. AlphaFold2 Models and Experimental Structures

A recent breakthrough in structural biology is based on computational approaches, which combine artificial intelligence and energy optimization. The AlphaFold2 neural network predicted the 3D structures of all proteins in humans and 20 model organisms [71,146]. The RoseTTAFold server [147] is another major resource based on artificial intelligence. Although models of transmembrane proteins are cautioned to be of limited precision, the superposition of the hNav1.5 cryo-EM structure (6lqa) with the AlphaFold2 structure (Figure 3) shows an impressive similarity of the transmembrane and extracellular segments. The AlphaFold2 structure also shows some structured segments in the cytoplasmic parts that are not resolved in the cryo-EM structure. Below we compare some crystal and cryo-EM structures with the respective AlphaFold2 models.

Superposition of five experimental structures of voltage-gated sodium channels (hNav1.2, hNav1.7, hNav1.4, and rNav1.5) with seven AlphaFold2 structures (hNav2.1, hNav1.4, hNav1.5, rNav1.5, mNav1.5, hNav1.4, and hNav1.9) shows similar folding and backbone conformations (Figure 4A,B). Structural deviations between the AlphaFold2 and experimental structures, on one hand, and between different experimental structures, on the other hand, are similar. A more detailed comparison is provided for Kv channels. Figure 4C shows the superimposed crystal structure of the Kv1.2–Kv2.1 channel with the AlphaFold2 structures of the voltage-gated potassium channels hKv1.2, hKv1.6, hKv2.1, and hKv3.1. The backbone conformations, positions, and orientation of the CA–CB bonds in these structures are very similar. RMS deviations (Å) of alpha carbons in the membrane segments of the Kv1.2 channel are as small as follows: S1, 1.2; S2, 1.1; S3, 1.5; S4, 0.73; S5, 0.46; P, 0.24; and S6, 0.53. The RMSD value for the P-helix is the smallest one because this segment was used for the 3D alignment. Thus, all details, including the open-gate conformation of the S6 bundle, are precisely predicted in the AlphaFold2 model. The AlphaFold2 models for the hKv1.6, hKv2.1, and hKv3.1 channels are also very close to the experimental structures. 

Figure 5A shows the experimental structures and AlphaFold2 models of two potassium channels: Kv1.2 (2R9R vs. P16389) and Kv7.1 (6uzz vs. P51787). Folding of the P-loops in the two channels is similar, but the folding of VSDs is rather different. The activation gate conformations (C-part of S6 and N-part of S5) also differ significantly. It is interesting, however, that the AlphaFold2 and experimental structures of each channel are similar. 

The next example includes VSD-IV of the hNav1.5 (Q14524) and hNav2.1 (Q01118) channels, where the CA–CB bonds of residues in sequentially matching positions of helices IVS1, IVS2 and IVS3 have similar orientations. However, there is a one-helical shift of helix IVS4 in the Nav2.1 channel towards the cytoplasm due to an additional helical turn between helix IVS4 and the linker-helix IVS4-S5 (Figure 5B). Since Nav2.1 (SCN7A) is not a voltage-gated channel, the atypical conformation of the S4 helix is not surprising. We are not aware of experimental structures of the Nav2.1 channel or other structures with atypical folding of the S4 helix.

Figure 5C shows an example of differences between P-loop channels. Although the P1 helices are the most structurally conserved elements of P-loop channels, the experimental structures demonstrate subtle, but notable differences between potassium and potassium-like glutamate receptor channels, on one hand, and sodium and calcium channels, on the other hand. In the latter family, the P-helices are about a half-turn more distant from the pore axis. The AlphaFold2 models readily reproduced this difference. Within each family, the pore helices of experimental and modeled structures are hardly distinguishable. 

The majority of channels with voltage-sensing domains have so-called swapped-domain architecture with a given VSD approaching the neighboring-domain quarter of the pore module (Figure 1C). However, non-swapped architecture is seen in some channels, e.g., hERG, K_Ca_1.1, and TRPV6 (5iwk). An alternative structure of the TRPV6 channel with resolved S4–S5 linkers (6e2f) shows the classical domain-swapping organization. AlphaFold2 structures of the K_Ca_1.1 and hERG channels have the non-swap domain architecture, whereas TRPV6 and other TRP channels have the swapped-domain architecture.

We further considered some sodium, calcium, and TRP channels with distortions caused by π-bulges. These channels have conserved asparagine residues in positions S6/56 (Figure 1D), which likely appeared in evolution as insertions [139] and induced π-bulges to preserve residue orientations in the S6 helices. The bulges are usually formed one turn upstream of the conserved asparagine, whose sidechain can donate an H-bond to the “bachelor” backbone carbonyl, and thus stabilize the bulge. The difference between S6 conformations in structures with and without π-bulges is shown in Figure 6A. The CA–CB bonds of residues in positions S6/46 and S6/56 are shown as sticks. Positions and orientations of CA–CB bonds in residues S6/46 are well conserved. However, CA–CB bonds of residues S6/56 are split due to the π-helix bulge at position S6/51 in TRPA1 (O75762) and TRPV3 (6lgp). AlphaFold2 structures of TRPM2 (O94759), TRPA1 (O75762), TRPV6 (Q9H1D0), TRPV1 (Q8NER1), and TRPV3 (Q8NET8) also have a π-bulge in S6.

Figure 6B,C shows intracellular views of the S6 bundle in eight sodium channels and six calcium channels. Folding of the transmembrane segments in the AlphaFold2 models is very similar to that in respective experimental structures. Particularly, residues in position S6/42 are well conserved in both experimental and AlphaFold2 structures. In contrast, residues in position S6/56 are split in two groups, depending on the presence or absence of π-bulges. The orientation of asparagines in position S6/56 varies between individual channels, between repeats of these channels, and between experimental structures and AlphaFold2 models. 

The experimental structures of the Nav1.2, Nav1.4, Nav1.5, and Nav1.7 channels have bulges in IS6 and IIIS6. Most of the AlphaFold2 models of channels hNav1.1, hNav1.2, hNav1.4, hNav1.5, and hNav1.9 have π-bulges in helices IS6, IIIS6, and IVS6, whereas the experimental structures lack a π-bulge in IVS6. AlphaFold2 models of the mNav1.5 and hNav1.9 channels have π-bulges in all of the four repeats, which match the π-bulge pattern in the experimental structure of NavPaS. Interestingly, the AlphaFold2 model of mNav1.5 differs from the rNv1.5 and hNav1.5 models, despite the sequences of these channels being very similar. Bulges in IS6 and IIIS6 are predicted for Nav2.1, which is not a voltage-gated channel. 

A similar situation is observed in calcium channels. The available experimental structures of these channels are split in several classes according to the presence or absence of π-helix bulges in the S6 helices [30]. The Class I structure has no π-bulges and Class II has π-bulges in repeats I, II, and III, whereas the Class III structure has bulges in repeats I and II. The AlphaFold2 model of the T-type calcium channel hCav3.1 reproduced π-bulges, which are seen in the experimental Class II structure. The AlphaFold2 model of Cav1.1 has a π-bulge only in repeat I. AlphaFold2 models of Cav1.3, Cav1.4, Cav2.1, and Cav2.2 have bulges in repeats I and III, which are often seen in experimental structures of sodium, but not calcium channels. The AlphaFold2 models of hCav3.2 and hCav3.3 have π-bulges in all four repeats.

Thus, patterns of π-helix bulges in the S6 helices are highly diverse in both the experimental structures and the AlphaFold2 models. The causes of the differences are unknown. Likely, there are alternative conformations with similar energies, and therefore transitions between the conformations are possible. Structural determinants underlying such transitions are unknown. This is an intriguing problem in the field of the structural biology of P-loop channels.

The AlphaFold2 publications mention a limited reliability of membrane protein models. However, the above comparison of the AlphaFold2 models and experimental structures of P-loop channels shows an impressive predictive power of the artificial neural network in this particular class of important membrane proteins. 

## 7. Perspectives

Further studies of P-loop channels in different states and in complexes with different ligands are necessary to address challenging problems involving the mechanisms of disease mutations. Since AlphaFold2 does not necessarily predict the consequences of missense mutations [148], such studies should involve a combination of experimental and theoretical approaches. P-loop channels are regulated by various auxiliary subunits and multiple cytoplasmic proteins, e.g., [149,150,151,152]. Some cryo-EM structures show complexes of channels with auxiliary subunits. However, the 3D structures of large cytoplasmic parts of many P-loop channels, which are targeted by various proteins, are not resolved in either cryo-EM structures or AlphFold2 models. Predicting the structures of P-loop channels with cytoplasmic proteins is of paramount importance for understanding the mechanisms by which disease mutations of the cytoplasmic proteins cause the ion channel dysfunction. Mutational studies, which reveal residues involved in protein–protein interactions, may provide important constraints to predict the protein–protein complexes using neural networks or physics-based protein–protein docking software. Another major problem is that the vast majority of experimental structures show P-loop channels in the energetically most-preferable states and neural-network software, which is trained on these structures, also predicts energetically preferable structures. Computational approaches may be used to transfer the energetically preferable experimental structures to low-populated, but functionally important states. Such models help to understand the mechanisms of disease mutations and ligand action [95,96,153]. Computational studies including molecular dynamic simulations and high-throughput ligands docking will benefit from the available experimental structures and neural-network-based models of P-loop channels. 

## Figures and Tables

**Figure 1 membranes-12-00229-f001:**
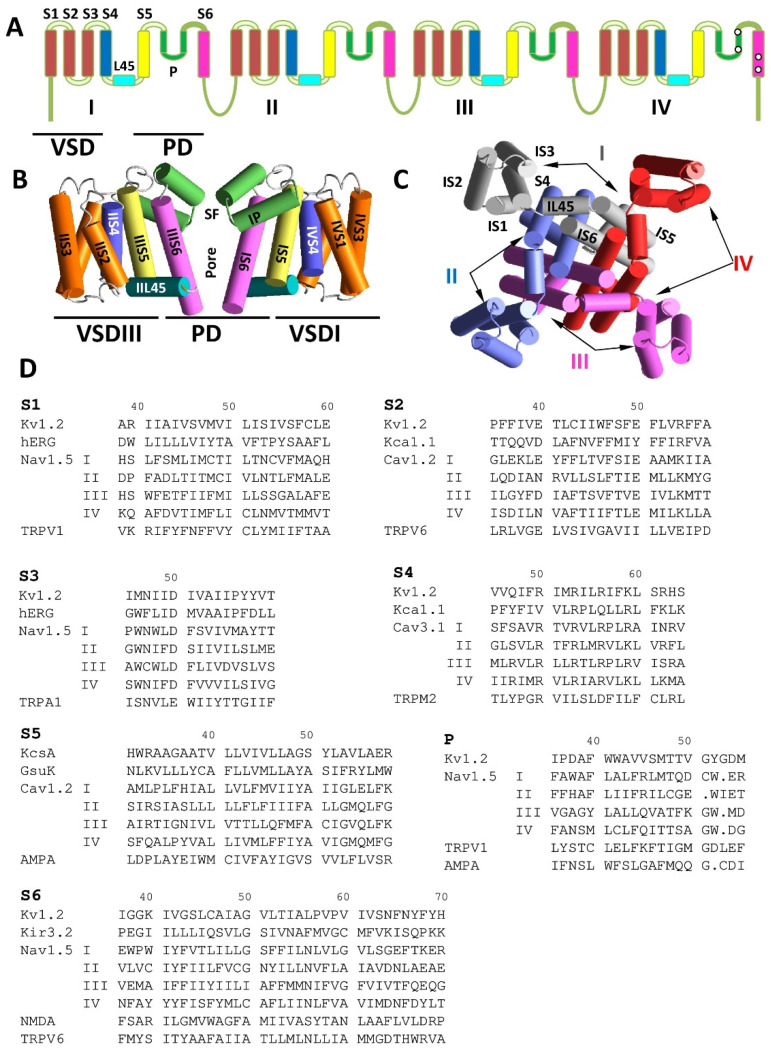
**Transmembrane topology and folding of some P-loop channels.** (**A**) Transmembrane topology of a eukaryotic sodium and calcium channel. Roman numerals indicate repeats in heterotetrameric channels. Voltage-sensing domains (VSD) and pore domain (PD) are marked. (**B**) Side view of the NavAb X-ray structure (3rvy). Only two repeats are shown for clarity. The segments are colored as in A. The P-loops have the selectivity filter (SF). (**C**) Intracellular view of the NavAb crystal structure with P-loops removed for clarity. Individual subunit repeats are shown with different colors. All four repeats contribute to the PD that surrounds the central pore, whereas the VSDs are localized peripherally. (**D**) Aligned sequences of helical segments some P-loop channels. The alignment is based on three-dimensional (3D) positions and orientations of residues in experimental structures.

**Figure 2 membranes-12-00229-f002:**
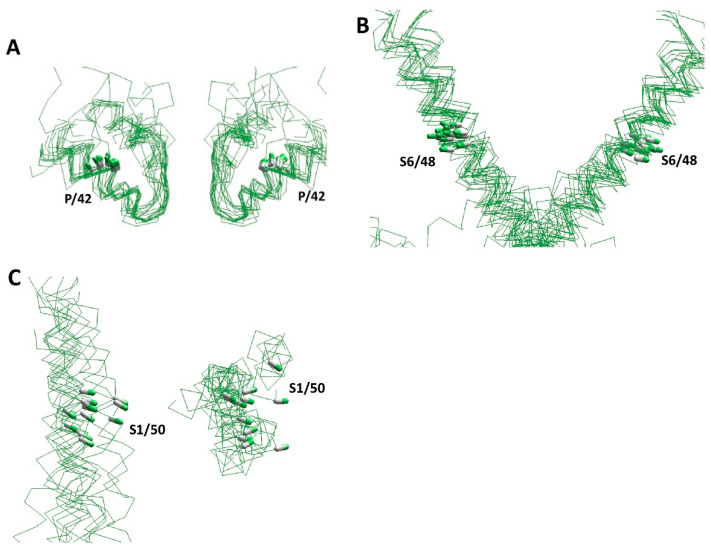
**Conserved folding of experimental structures of P-loop channels.** Individual segments are shown for the 3D-aligned structures of the following channels: KcsA (1bl8), Kv1.2–Kv2.1 (2r9r), MlotiK1 (3beh), TRPV1 (3j5r), TRPA1 (3j9p), NavAb (3rvy), GsuK (4gx5), Kir3.2 (4kfm), Slo (5tj6), hERG (5va2), NavPaS (5x0m), rbCav1.1 (6byo), TPC1 (6c96), NMDA (6cna), AMPA (6dm0), Cav3.1 (6kzp), TRPM2 (6mj2), Nav1.5 (6uz3), and TRPV6 (6e2g). CA–CB bonds of residues in matching positions of the sequence alignments (Figure 1D) are shown by sticks. (**A**) Side view of P-loops. (**B**) Side view of S6 helices. Despite significant sequential, functional, and structural diversity of the channels, the positions and orientations of the CA–CB bonds are very similar in P-loops and S6 helices. (**C**) Side (left panel) and extracellular (right panel) views of S1 helices. Despite the similarity of VSD backbones in the structures, which were 3D-aligned by minimizing RMS deviations of P-helices, is less obvious, the positions and orientations of the CA–CB bonds of residues S1/50 are comparable.

**Figure 3 membranes-12-00229-f003:**
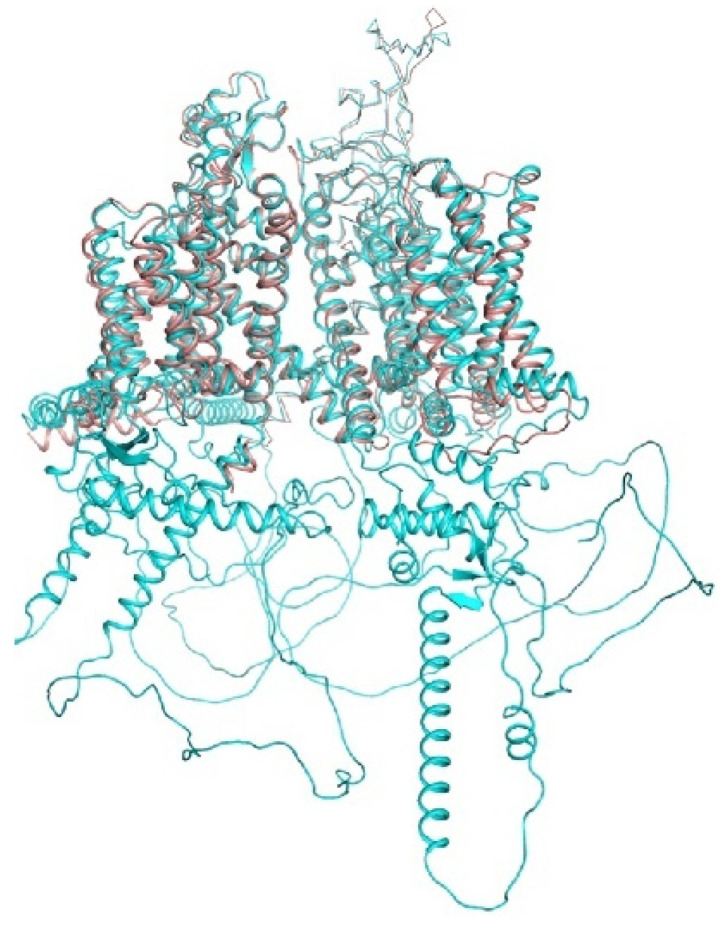
**hNav1.5 channel.** In the AlphaFold2 structure (Q14524, cyan) and cryo-EM structure (6LQA; brown), transmembrane and extracellular segments are similar. However, the AIphaFold2 structure also shows some structured segments in the cytoplasmic parts, which are not resolved in the cryo-EM structure.

**Figure 4 membranes-12-00229-f004:**
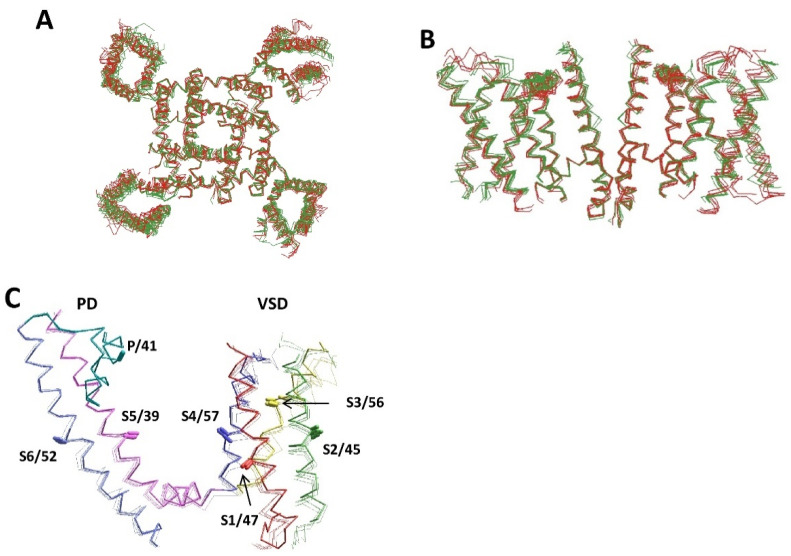
**Similarity of experimental and AlphaFold2 structures of voltage-gated potassium and sodium channels.** (**A**,**B**) Intracellular (**A**) and intra-membrane (**B**) views of the experimental (green) and AlphaFold2 (red) structures of voltage-gated sodium channels. Shown are the experimental structures of hNav1.2 (6j8e), hNav1.7 (6j8j), hNav1.4 (6agf), and rNav1.5 (6uz3), and the AlphaFold2 structures of hNav2.1 (Q01118), hNav1.4 (P35499), hNav1.5 (Q14524), rNav1.5 (P15389), mNav1.5 (Q9JJV9), hNav1.4 (P35499), and hNav1.9 (Q9UI33). For clarity, only repeats II and IV are shown in (**B**,**C**). (**C**) Crystal structure of the Kv1.2 channel (DPB ID: 2R9R) superimposed with the AlphaFold2 structures of hKv1.2 (P16389), hKv1.6 (P17658), hKv2.1 (Q14721), and hKv3.1 (P48547). Helices S1, S2, S3, S4, S5, and S6 are red, green, yellow, blue, magenta, and violet, respectively. P-loops are cyan. The CA–CB bonds of residues in matching positions are shown by sticks and labeled as in Figure 1D. The AlphaFold2 and experimental structures are very similar.

**Figure 5 membranes-12-00229-f005:**
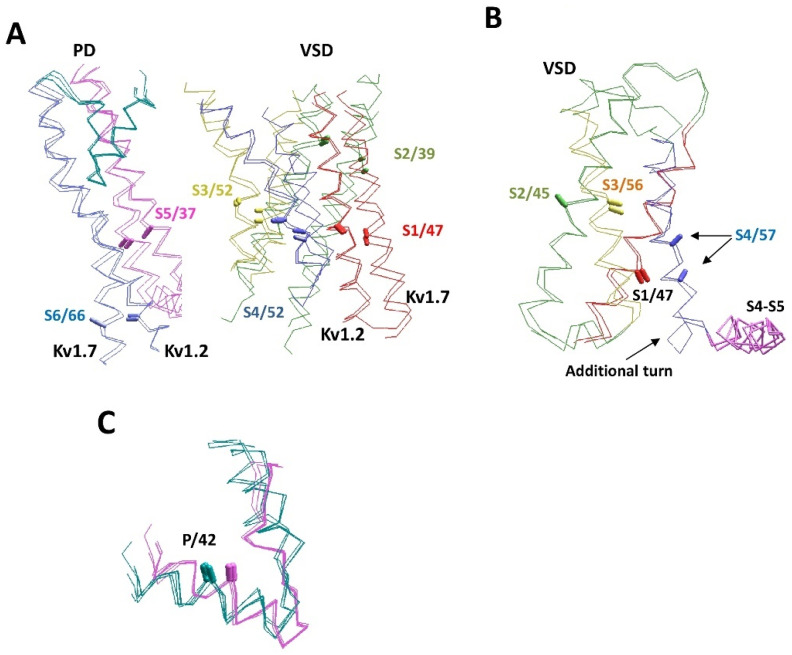
**Structural peculiarities of voltage-gated P-loop channels.** (**A**) The superimposition of experimental and AlphaFold2 structures of the Kv1.2 (2R9R and P16389) and Kv7.1 (6uzz and P51787) channels. Helices S1, S2, S3, S4, S5, and S6 are red, green, yellow, blue, magenta, and violet, respectively. P-loops are cyan. The CA–CB bonds of residues in matching positions are shown by sticks and labeled as in Figure 1D. Structures Kv1.2 and Kv7.1 are significantly different, but the AlphaFold2 and experimental structures of each channel are similar. (**B**) VSD-IV in channels hNav1.5 (Q14524) and hNav2.1 (Q01118). The CA–CB bonds of homologous residues in transmembrane segments match well, except for helix IVS4. In Nav2.1, the helix is shifted towards the cytoplasm due to the appearance of an additional helical turn between helix IVS4 and linker-helix S4–S5. (**C**) P-loops in the experimental and AlphaFold2 structures of potassium channels (cyan) are significantly different from sodium and calcium channels (magenta), but within each subfamily, the experimental and AlphaFold2 structures are similar. Shown are channels Kv1.2-2.1 (2r9r), hKv1.5 (6uzz), rbCav1.1 (5gjv), hNav1.4 (6agf), hKv7.1 (P51787), hKv2.1 (Q14721), hKv1.2 (P16389), hKv1.6 (P17658), hKv3.1 (P48547), hCav3.1 (O43497), hNav1.2 (Q99250), and hNav1.5 (Q14524).

**Figure 6 membranes-12-00229-f006:**
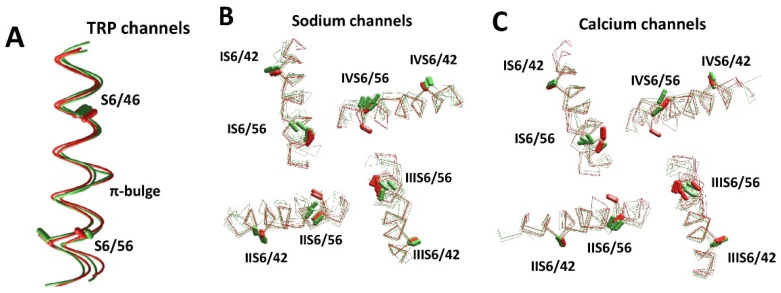
**Diverse conformations of S6 helices in experimental and AlphaFold2 structures.** (**A**) S6 helices in TRP channels. Experimental structures of TRPV6 (5iwk) and TRPV3 (6lgp) are green. AlphaFold2 structures of TRPV2 (Q9Y5S1) and TRPA1 (O75762) are red. Positions and orientations of the CA–CB bonds of residues S6/46 are well conserved. In contrast, the CA–CB bonds of residues S6/56 are split due to the π-helix bulge at position S6/51 in the TRPA1 (O75762) and TRPV3 (6lgp) structures. (**B**,**C**) S6 helices in eukaryotic sodium (**B**) and calcium (**C**) channels. Experimental and AlphaFold2 structures are green and red, respectively. (**B**) Sodium channels hNav1.2 (6j8e), hNav1.7 (6j8j, 6n4q), hNav1.4 (6agf/P35499), rNav1.5 (6uz3/P15389), hNav2.1 (Q01118), hNav1.5 (Q14524), mNav1.5 (Q9JJV9), and hNav1.9 (Q9UI33). (**C**) Calcium channels hCav3.1 (6kzo/O43497), rbCav1.1 (6jpa, 6jp5, 6byo and 5gjv), hCav1.1 (Q13698), hCav1.4 (O60840), hCav1.3 (Q01668), and hCav3.2 (O95180). The CA–CB bonds of residues S6/42 and S6/65 (Figure 1D) are shown by sticks. The positions and orientations of the CA–CB bonds in position S6/42 are similar in the experimental and AlphaFold2 structures. However, due to the different patterns of π-helical bulges, the orientation of asparagines in position S6/56 varies between repeats of individual channels and between different structures.

**Table 1 membranes-12-00229-t001:** Ion channel structures discussed in this work.

Channel	PDB ID	Ref.	AF ^a^	Channel	PDB ID	Ref.	AF ^a^
**Potassium**				**Calcium**			
KcsA	1bl8	[26]		rbCav1.1	5gjv	[30]	
MthK	6u6e	[31]			6jpa	[32]	
MlotiK1	3beh	[33]			6jpb	[32]	
Kv1.2/Kv2.1	2r9r	[28]			7jpk	[34]	
hKv1.2			P16389		7jpw	[34]	
hKv1.6			P17658		6jp5	[32]	
hKv2.1			Q14721	hCav1.1			Q13698
hKv3.1			P48547	hCav1.3			Q01668
hKv7.1	6uzz	[35]	P51787	hCav3.1	6kzo	[36]	O43497
hERG	5va2	[37]			6kzp	[36]	
Kir3.2	4kfm	[38]		hCav3.2			O95180
GsuK	4gx5	[39]		hCav1.4			O60840
Slo	5tj6	[40]					
**Sodium**				**iGluR**			
NavAb	3rvy	[41]		AMPA	6dm0	[42]	
	5vb2	[43]		NMDA	6cna	[44]	
	5vb8	[43]					
	6p6x	[45]		**TRP**			
	6pwp	[45]		TRPA1	3j9p	[46]	O75762
NavMs	4f4l	[47]		TRPV1	3j5r	[48]	Q8NER1
NavRh	4dxw	[49]		TRPV2	6oo7	[50]	Q9Y5S1
NavPaS	6a95	[51]		TRPV3	6dvy	[52]	Q8NET8
Nav1.2	6j8e	[53]	Q01118		6mhw	[54]	
hNav1.4	6agf	[55]	P35499		6mho	[54]	
EeNav1.4	5xsy	[56]			6pvl	[57]	
rNav1.5	6uz3	[58]	P15389		6dvw	[52]	
	7fbs	[59]			6uw9	[60]	
	7k18	[61]			6lgp	[62]	
hNav1.5	6lqa	[63]	Q14524		6uw4	[60]	
hNav1.7	6j8j	[64]			6mhs	[54]	
	6n4r	[65]		TRPV6	5iwk	[66]	
hNav1.9			Q9UI33		6e2g	[67]	Q9H1D0
hNav2.1			Q01118	TRPM2	6mj2	[68]	O94759
TPC1	6c96	[69]		TRPM6	6Bo9	[70]	

^a^ UniProt accession code of neural network-predicted structures of proteins deposited in the AlphaFold protein structure database https://alphafold.ebi.ac.uk, accessed on 25 December 2021 [71,72].

## Data Availability

Not applicable.

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
