# Peer review of "P-Loop Channels: Experimental Structures, and Physics-Based and Neural Networks-Based Models"

_membranes, 2022, doi:10.3390/membranes12020229_

Round 1
Reviewer 1 Report
In this manuscript the authors reviewed the literature describes the P-LOOP channels as a experimental structures and computational models.
This review is interesting, nevertheless needs some improvements before publishing:
Special points:
Lines 28-45: please add multiple references at the end of each these sentences.
Lines 69-75: please add multiple references at the end of each these sentences.
Lines 89-92: please add multiple references at the end of this sentence.
Lines 125-139: please add multiple references at the end of each these sentences.
Lines 137-141: please add multiple references at the end of each these sentences.
Lines 151-152: please add multiple references at the end of this sentence.
Lines 165-168: please add multiple references at the end of each these sentences.
Lines 183-188: please add multiple references at the end of each these sentences.
Lines 195-196: please add multiple references at the end of this sentence.
Lines 201-202: please add multiple references at the end of this sentence.
Lines 215-217: please add multiple references at the end of this sentence.
Lines 224-225: please add multiple references at the end of this sentence.
Lines 231-234: please add multiple references at the end of each these sentences.
Lines 247-253: please add multiple references at the end of each these sentences.
Lines 257-261: please add multiple references at the end of each these sentences.
Lines 271-274: please add multiple references at the end of each these sentences.
Lines 282-287: please add multiple references at the end of each these sentences.
Lines 297-298: please add multiple references at the end of this sentence.
Lines 300-301: please add multiple references at the end of this sentence.
Lines 305-306: please add multiple references at the end of this sentence.
Lines 312-324: please add multiple references at the end of each these sentences.
Lines 332-336: please add multiple references at the end of each these sentences.
Lines 354-367: please add multiple references at the end of each these sentences.
Lines 369-380: please add multiple references at the end of each these sentences.
Lines 398-414: please add multiple references at the end of each these sentences.
Lines 433-451: please add multiple references at the end of each these sentences.
Author Response
Dear Reviewer,
Thank you very much for reviewing our manuscript. In the revised version, we have made corrections to the text, figures and references in accordance with your comments and the comments of another reviewer. Below, please find our answers to your specific comments.
Sincerely,
Denis Tikhonov and Boris Zhorov.
Comment. This review is interesting, nevertheless needs some improvements before publishing:
Special points:
Lines 28-45: please add multiple references at the end of each these sentences.
Lines 69-75: please add multiple references at the end of each these sentences.
Lines 89-92: please add multiple references at the end of this sentence.
Lines 125-139: please add multiple references at the end of each these sentences.
Lines 137-141: please add multiple references at the end of each these sentences.
Lines 151-152: please add multiple references at the end of this sentence.
Lines 165-168: please add multiple references at the end of each these sentences.
Lines 183-188: please add multiple references at the end of each these sentences.
Lines 195-196: please add multiple references at the end of this sentence.
Lines 201-202: please add multiple references at the end of this sentence.
Lines 215-217: please add multiple references at the end of this sentence.
Lines 224-225: please add multiple references at the end of this sentence.
Lines 231-234: please add multiple references at the end of each these sentences.
Lines 247-253: please add multiple references at the end of each these sentences.
Lines 257-261: please add multiple references at the end of each these sentences.
Lines 271-274: please add multiple references at the end of each these sentences.
Lines 282-287: please add multiple references at the end of each these sentences.
Lines 297-298: please add multiple references at the end of this sentence.
Lines 300-301: please add multiple references at the end of this sentence.
Lines 305-306: please add multiple references at the end of this sentence.
Lines 312-324: please add multiple references at the end of each these sentences.
Lines 332-336: please add multiple references at the end of each these sentences.
Lines 354-367: please add multiple references at the end of each these sentences.
Lines 369-380: please add multiple references at the end of each these sentences.
Lines 398-414: please add multiple references at the end of each these sentences.
Lines 433-451: please add multiple references at the end of each these sentences.
Response. We introduce additional references in the text. We also compiled Table 1 that shows PDB indexes of all the structures mentioned in the text, respective references, as well as AlphaFold2 indexes. Instead of placing references in multiple positions in the text, we assembled the references to the structural papers in this Table.
Reviewer 2 Report
In the Review “P-Loop Channels: Experimental Structures and Computational Models” by Denis B. Tikhonov and Boris S. Zhorov undertook a very interesting subject, the molecular structure of P-Loop channels. They analyze the structure of these channels based on experimental and machine learning approaches which do not always give completely the same and compatible information, especially when it comes to channels containing π-helical bulges in the inner helices. The described problem seems to be important for the understanding of P-loop channels activity, possibilities of their modulation. The article is high of importance, however, in many places is hard to understand by not specialists in crystallography. In the reviewer opinion, the authors should adjust the manuscript to a wider audience.
Specific comments:
1)The title of the manuscript does not specify precisely the problem undertaken and analyzed by authors in the Review and should be improved. It does not follow from the title that the comparison of models AlphaFold to experimental was undertaken in the article. Please change the title to make it specific and relevant to the article message.
2)The Article message is not clear throughout the manuscript. Only verses 448-453 summarizing the article suggest the aim undertaken in the Review. Please consider reorganizing the information you describe. You may e.g. apply MDPI suggestions and introduce sections: Introduction, Discussion, Conclusions, and Future Directions (https://www.mdpi.com/about/article_types) which should help the readers understand the main message of the Review Article.
3)There is no literature on the information described in verses 28-45. Please add the appropriate reference.
4)Please consider moving the second paragraph from the section “1. Architecture of P-loop Channels” about the types of P-loop channels to a different part of the article, e.g. to the introduction section, and describe deeper the systematics of these channels or send the reader to appropriate literature. Very helpful would be the table where the reader could find all P-loop channel members described in the manuscript with appropriate literature references to detailed information about these channels.
5)Please improve Figures and figure captions so they are easy to understand without looking at the body text and also could help better understand the body text of the manuscript. In the current version, readers cannot find, e.g. wherein Fig. 1. is the architecture of P-channels described in lines 33-36, I mean it is not indicated in the figure where are “P1 helices”, “C ends”, “narrowing pore” etc. There is no explanation in Fig.1A the meaning “VSD”, ”PD”. Moreover, the rationale of the illustration in The Fig. 1D, the sequence alignment of S1-S6 and P segments for different P-channels is not clear. Please correct.
6)Lines 294-295 but also lines 451-453 please specify which kind of studies you expect would be necessary and how they help to sort described problem out. Could you suggest some?
7)I recommend correcting the usage of the abbreviations throughout the article. Each abbreviation should be explained when it is introduced in the body text and then should be used consistently. In the current version, abbreviations are not explained when they are used for the first time, sometimes they are not explained et all. Please correct.
8)A list of abbreviations is very helpful for readers to understand the text. However, in the current version, the list is not complete. Please complete the list of abbreviations in the manuscript.
9)Fig 6A is not cited in the text. Please correct.
10)There is no information about the source of material presented in Figures 1-6. Even if presented figures are taken from own published writing, the source of presented in figures material has to be cited and appropriate information about obtained permission should be indicated. Please complete.
Author Response
Dear Reviewer,
Thank you very much for reviewing our manuscript. In the revised version, we have made corrections to the text, figures and references in accordance with your comments and the comments of another reviewer. Below, please find our answers to your specific comments.
Sincerely,
Denis Tikhonov and Boris Zhorov.
Comment. In the Review “P-Loop Channels: Experimental Structures and Computational Models” by Denis B. Tikhonov and Boris S. Zhorov undertook a very interesting subject, the molecular structure of P-Loop channels. They analyze the structure of these channels based on experimental and machine learning approaches which do not always give completely the same and compatible information, especially when it comes to channels containing π-helical bulges in the inner helices. The described problem seems to be important for the understanding of P-loop channels activity, possibilities of their modulation. The article is high of importance, however, in many places is hard to understand by not specialists in crystallography. In the reviewer opinion, the authors should adjust the manuscript to a wider audience.
Specific comments:
Comment 1) The title of the manuscript does not specify precisely the problem undertaken and analyzed by authors in the Review and should be improved. It does not follow from the title that the comparison of models AlphaFold to experimental was undertaken in the article. Please change the title to make it specific and relevant to the article message.
Response: The title is changed
Comment 2)The Article message is not clear throughout the manuscript. Only verses 448-453 summarizing the article suggest the aim undertaken in the Review. Please consider reorganizing the information you describe. You may e.g. apply MDPI suggestions and introduce sections: Introduction, Discussion, Conclusions, and Future Directions (https://www.mdpi.com/about/article_types) which should help the readers understand the main message of the Review Article.
Response: We renamed section 1 that now reads: “1. Introduction”. We clarified aims of the Review at the end of this section. We also introduced section 7 “Perspectives. However, we do not think that section “Discussion” is appropriate for our Review article.
Comment 3) There is no literature on the information described in verses 28-45. Please add the appropriate reference.
Response. Corrected. We introduced references to recent reviews on specific channel families.
Comment 4)Please consider moving the second paragraph from the section “1. Architecture of P-loop Channels” about the types of P-loop channels to a different part of the article, e.g. to the introduction section, and describe deeper the systematics of these channels or send the reader to appropriate literature. Very helpful would be the table where the reader could find all P-loop channel members described in the manuscript with appropriate literature references to detailed information about these channels.
Response. We renamed section 1 which now reads 1. Introduction. We also compiled Table 1 names of all channels mentioned in the paper and respective PDB IDs and references.
Comment 5)Please improve Figures and figure captions so they are easy to understand without looking at the body text and also could help better understand the body text of the manuscript. In the current version, readers cannot find, e.g. wherein Fig. 1. is the architecture of P-channels described in lines 33-36, I mean it is not indicated in the figure where are “P1 helices”, “C ends”, “narrowing pore” etc. There is no explanation in Fig.1A the meaning “VSD”, ”PD”.
Response. Figure 1 and its legend are corrected.
Comment. Moreover, the rationale of the illustration in The Fig. 1D, the sequence alignment of S1-S6 and P segments for different P-channels is not clear. Please correct.
Response. Our sequence alignment shown in Fig. 2D does not necessarily match the alignment proposed by web-servers (e.g. Clustal Omega). Sometimes our alignment if not obvious, but it ensures that residues in sequentially matching position have the sane positions/orientations in the aligned 3D structures. We have mentioned this in the revised version. The text and figures many times address specific positions in the alignment.
Comment 6)Lines 294-295 but also lines 451-453 please specify which kind of studies you expect would be necessary and how they help to sort described problem out. Could you suggest some?
Response. We remove these sentences. We also added section “7. Perspectives” in which we describe our view on further studies.
Comment 7)I recommend correcting the usage of the abbreviations throughout the article. Each abbreviation should be explained when it is introduced in the body text and then should be used consistently. In the current version, abbreviations are not explained when they are used for the first time, sometimes they are not explained et all. Please correct.
Response. Corrected.
Comment 8)A list of abbreviations is very helpful for readers to understand the text. However, in the current version, the list is not complete. Please complete the list of abbreviations in the manuscript.
Response. Corrected.
Comment 9) Fig 6A is not cited in the text. Please correct.
Response. This Figure is now mentioned in the text.
Comment 10)There is no information about the source of material presented in Figures 1-6. Even if presented figures are taken from own published writing, the source of presented in figures material has to be cited and appropriate information about obtained permission should be indicated. Please complete.
Response. The original structures are downloaded from the Protein Data Bank (www.rcsb.org) and the AlphaFold2 databank (www.alphafold.ebi.ac.uk/download). In the revised version it is mentioned in the section “Comparative Structural Analysis of P-loop Channels”.
Reviewer 3 Report
the work presented discussing P-Loop Channels: Experimental Structures and Computational Models by Tikhonov S. Zhorov aims to describe the structure-function relation of the superfamily of P-loop channels seems to be fitting more a textbook than a review article.
the write-up lacks proper flow as there abstract and the different sections do not clearly describe what are the aims of this review?
This is slightly mentioned in the introduction "this review we describe crystal and cryo-EM structures and physics-based models 70 of various P-loop channels and their complexes with drug and toxins." however, this is mildly touched upon in the different sections and the review goes deep in biochemistry rather than discussing the implication sof p-loops in toxis, effect of mutations in neurodeerative disease, effect of difference and similarities of the different channels of the P-loop channels.
Thus, this work needs to be rewritten again in a more didactic approach that has translational implication and with a focus to fill a gap of knowledge, in its current situation it reads like a biochemistry chapter in a book.
minor comments:
1-the writing uses acronyms without defining them such as the TRP channels
2-repetitious sentences: the last two decades (Table 1 and refs therein).
3- wrong sentence structure: whose crystal structure was obtained the first eukaryotic P-loop potassium
Author Response
Dear Reviewer,
Thank you very much for reviewing our manuscript. In the revised version, we have made corrections to the text in accordance with your comments. Below, please find our answers to your specific comments.
Sincerely,
Denis Tikhonov and Boris Zhorov.
Comment. the work presented discussing P-Loop Channels: Experimental Structures and Computational Models by Tikhonov S. Zhorov aims to describe the structure-function relation of the superfamily of P-loop channels seems to be fitting more a textbook than a review article.
the write-up lacks proper flow as there abstract and the different sections do not clearly describe what are the aims of this review?
This is slightly mentioned in the introduction "this review we describe crystal and cryo-EM structures and physics-based models 70 of various P-loop channels and their complexes with drug and toxins." however, this is mildly touched upon in the different sections and the review goes deep in biochemistry rather than discussing the implication sof p-loops in toxis, effect of mutations in neurodeerative disease, effect of difference and similarities of the different channels of the P-loop channels.
Response. In the revised version we provide additional paragraph that describe our motivation and the aim of the present review. The problems mentioned by the Reviewer are addressed in numerous papers, and we provided citations to several excellent reviews. However, modern literature generally lacks comparative analysis of different members of P-loop channels. Structural and functional insights that can arise from the similar pore-forming domain organization in all these channels remain underestimated. Thus, the review is aimed to fill this gap. Our work is addressed to researches interested in the structural organization of various members of the superfamily and may get new insights from the comparative view.
Thus, this work needs to be rewritten again in a more didactic approach that has translational implication and with a focus to fill a gap of knowledge, in its current situation it reads like a biochemistry chapter in a book.
Response. Please see our answer to the above comment.
minor comments:
1-the writing uses acronyms without defining them such as the TRP channels
Response. Corrected. Other abbreviations are also defined (marked in the revised manuscript).
2-repetitious sentences: the last two decades (Table 1 and refs therein).
Response. Rephrased: “Over the past two decades, numerous experimental structures of various P-loop channels have been published (Table 1 and references therein).”
3- wrong sentence structure: whose crystal structure was obtained the first eukaryotic P-loop potassium
Response. Changed to “the first eukaryotic P-loop channel whose crystal structure was obtained with the resolution below 2.5 Å”
Round 2
Reviewer 1 Report
Once again, please do all your corrections according all my previously proposals:
Special points:
Lines 28-45: please add multiple references at the end of each these sentences.
Lines 69-75: please add multiple references at the end of each these sentences.
Lines 89-92: please add multiple references at the end of this sentence.
Lines 125-139: please add multiple references at the end of each these sentences.
Lines 137-141: please add multiple references at the end of each these sentences.
Lines 151-152: please add multiple references at the end of this sentence.
Lines 165-168: please add multiple references at the end of each these sentences.
Lines 183-188: please add multiple references at the end of each these sentences.
Lines 195-196: please add multiple references at the end of this sentence.
Lines 201-202: please add multiple references at the end of this sentence.
Lines 215-217: please add multiple references at the end of this sentence.
Lines 224-225: please add multiple references at the end of this sentence.
Lines 231-234: please add multiple references at the end of each these sentences.
Lines 247-253: please add multiple references at the end of each these sentences.
Lines 257-261: please add multiple references at the end of each these sentences.
Lines 271-274: please add multiple references at the end of each these sentences.
Lines 282-287: please add multiple references at the end of each these sentences.
Lines 297-298: please add multiple references at the end of this sentence.
Lines 300-301: please add multiple references at the end of this sentence.
Lines 305-306: please add multiple references at the end of this sentence.
Lines 312-324: please add multiple references at the end of each these sentences.
Lines 332-336: please add multiple references at the end of each these sentences.
Lines 354-367: please add multiple references at the end of each these sentences.
Lines 369-380: please add multiple references at the end of each these sentences.
Lines 398-414: please add multiple references at the end of each these sentences.
Lines 433-451: please add multiple references at the end of each these sentences.
Author Response
Dear Reviewer,
In the revised version we made modifications in accordance to your comments. They are marked in the text. Below. Please find our responses to your specific comments.
Sincerely,
Denis Tikhonov and Boris Zhorov.
Lines 28-45: please add multiple references at the end of each these sentences.
Response: A total of 19 refs are added to these lines. It is impossible to provide multiple citations at the end of each sentence in these and later mentioned lines: such refs would be redundant and some statements do need refs at al.
Lines 69-75: please add multiple references at the end of each these sentences.
Response: A total of 6 refs are added to these lines.
Lines 89-92: please add multiple references at the end of this sentence.
Response: Three refs are added
Lines 125-139: please add multiple references at the end of each these sentences.
Response: We added a ref to Table 1 with multiple refs. The rest of the para describes structural features of P-loop channels shown in Figure 2. The Figure legend contains PDB IDs, refs to which are provided in Table 1.
Lines 137-141: please add multiple references at the end of each these sentences.
Response: Five refs are added
Lines 151-152: please add multiple references at the end of this sentence.
Response: This is a general statement. Refs are provided in the subsequent sentences.
Lines 165-168: please add multiple references at the end of each these sentences.
Response: This is a general statement. Refs are provided in the subsequent sentences.
Lines 183-188: please add multiple references at the end of each these sentences.
Response: This is a general statement. Refs are provided in the subsequent sentences.
Lines 195-196: please add multiple references at the end of this sentence.
Response: Ref to a review is added.
Lines 201-202: please add multiple references at the end of this sentence.
Response: This is a general statement. Refs are provided in the subsequent sentences.
Lines 215-217: please add multiple references at the end of this sentence.
Response: This is a general statement. Refs are provided in the subsequent sentences.
Lines 224-225: please add multiple references at the end of this sentence.
Response: Ref to a review is added.
Lines 231-234: please add multiple references at the end of each these sentences.
Response: This is a general statement. Refs are provided in the subsequent sentences.
Lines 247-253: please add multiple references at the end of each these sentences.
Response: References to multiple structures of channels with and without π-bulges are provided in Table 1. These structures are discussed in the subsequent sentences of this section and specific refs can also be found in Table 1
Lines 257-261: please add multiple references at the end of each these sentences.
Response: A ref is provided.
Lines 271-274: please add multiple references at the end of each these sentences.
Response: These lines contain multiple PDB IDs and respective refs can be found in Table 1
Lines 282-287: please add multiple references at the end of each these sentences.
Response: This is a conclusion. Refs are provided in the preceding sentences.
Lines 297-298: please add multiple references at the end of this sentence.
Response: This is a general statement. Refs are provided in the subsequent sentences.
Lines 300-301: please add multiple references at the end of this sentence.
Response: Refs to RoseTTAFold is provided in these lines.
Lines 305-306: please add multiple references at the end of this sentence.
Response: Respective PDB and AlphaFold2 indexes are provided in the legend to Fig. 3. Refs to all PDB structures, which are mentioned in the text, are given in Table 1. All AlphaFold2 structures are deposited the databank refs to which are provided at the legend to Table 1.
Lines 312-324: please add multiple references at the end of each these sentences.
Response: These lines compare PDB and AlphaFolf2 structures. Refs to all these structures are provided in Table 1
Lines 332-336: please add multiple references at the end of each these sentences.
Response: All AlphaFold2 structures are deposited the databank refs to which are provided at the legend to Table 1.
Lines 354-367: please add multiple references at the end of each these sentences.
Response: These lines compare PDB and AlphaFolf2 structures. Refs to all these structures are provided in Table 1
Lines 369-380: please add multiple references at the end of each these sentences.
Response: These lines compare PDB and AlphaFolf2 structures. Refs to all these structures are provided in Table 1
Lines 398-414: please add multiple references at the end of each these sentences.
Response: These lines compare PDB and AlphaFolf2 structures. Refs to all these structures are provided in Table 1
Lines 433-451: please add multiple references at the end of each these sentences.
Response: These lines compare PDB and AlphaFolf2 structures. Refs to all these structures are provided in Table 1
Reviewer 3 Report
accept
Round 3
Reviewer 1 Report
Thank you for all your corrections.
Author Response
Thank you again for the reviewing our manuscript.
Sincerely,
Denis Tikhonov and Boris Zhorov